# Causal Class Activation Maps for Weakly-Supervised Semantic Segmentation

**Yiping Wang**[1]

[1]Computer Science Dept., University of Waterloo, Waterloo, Ontario, Canada

## Abstract

Weakly-supervised semantic segmentation (WSSS) has emerged in recent years due to its appealing requirements for training data, i.e., with only image-level labels available as supervision. Most existing WSSS methods exploit the class activation maps (CAMs) as the seeds and generate the pseudo-pixel-level ground truth to train a segmentation network. In this work, we introduce a causal inference framework to ameliorate the quality of CAMs, conducing to the performance raise of existing WSSS algorithms that rely on CAMs. Our motivation is to deconfound a set of class-specific latent *confounders* in a dataset, which are the potential cause of low-quality and poorly-localized CAMs. Due to the unobservable nature of the confounders, we present the utilization of *front-door adjustment* for causal intervention to deconfound a classification neural network, without presuming and estimating the confounders explicitly. Our proposed algorithm, Causal CAM ($C^2AM$), outperformed the prior causal framework for WSSS Zhang et al. [2020] by a large margin, without any additional parameters, network architecture modification, or manipulation of images, and only needs to add one more line of code in a standard classifier training loop. Furthermore, we provide an optimization interpretation of the front-door adjustment for training a classifier to explain the improvements by $C^2AM$. We evaluated $C^2AM$ on PASCAL VOC 2012 and achieved mIoU 69.6% of pseudo-mask generation on the training set, and mIoU 67.5% and 67.7% on validation and test set after training DeepLabV2 on the pseudo-masks. Our implementation and model weights for reproducibility are released at `https://github.com/yiping-wang/c2am`

## 1 INTRODUCTION

Semantic segmentation is the task of classifying each pixel of an image into its corresponding semantic class Kirillov et al. [2019]. It is a core and fundamental building block for many visual computing applications, such as scene understanding Hofmarcher et al. [2019] and biomedical image analysis Havaei et al. [2017]. Previously, training deep learning models for semantic segmentation requires pixel-level annotations Long et al. [2015], Ronneberger et al. [2015], Chen et al. [2018], Minaee et al. [2021], which are expensive and laborious to obtain, e.g., annotating a $500 \times 500$ natural image with pixel-level ground truth for an object can easily take ten times longer than creating a bounding box around it Everingham et al. [2015]. In contrast, image-level labels are the easiest and cheapest to collect, which merely take almost one second per object-category Papadopoulos et al. [2014]. With the massive amount of image data available nowadays, weakly-supervised learning for semantic segmentation has been gaining attention for its "weak" requirements for labels of training data, and "weak" emphasizes the cheaper labelling cost at image-level Araslanov and Roth [2020], Wang et al. [2020].

Class Activation Maps (CAMs) Zhou et al. [2016], Selvaraju et al. [2017] has been a popular and powerful starting point, or *seed*, for most weakly-supervised semantic segmentation (WSSS) algorithms Kolesnikov and Lampert [2016], Araslanov and Roth [2020], Ahn et al. [2019], Wang et al. [2020], Ahn and Kwak [2018], Chen et al. [2022]. CAMs localize the most discriminative, albeit coarse and incomplete, regions of a semantic class in an image as the *seed* areas that are further exploited and expanded to obtain the pixel-level pseudo-masks Huang et al. [2018], Wei et al. [2018], which are treated as the pseudo-ground-truth for training a standard pixel-level supervised semantic segmentation algorithm Araslanov and Roth [2020], Zhang et al. [2021].

Despite the successful applications of CAMs for WSSS, CAMs are generated from a classification network, and

*Accepted for the 38th Conference on Uncertainty in Artificial Intelligence* (UAI 2022).

a classification network does not always learn the *causal features* that are robust in any confounding context, e.g., the foreground object features are invariant in any different background context Wang et al. [2021a]. This happens as classification does not necessitate a precise localization of the objects, the network could take advantage of the spurious correlation in the confounding context as long as it benefits the prediction when the training and testing data are i.i.d. Unfortunately, enlarging the scale of the datasets won't alleviate this bias Yang et al. [2021], as such biases are embedded in the nature of data, as indicated in Zipf's law Reed [2001]. Indeed, "yellow banana" occurs more often than "green banana" in reality. Thus, such spurious correlations and bias learned in a classification network could pose an inferior quality of CAMs, and lead to poor performance of WSSS algorithms that are based on CAMs. The fundamental solution to learn the robust causal features is by *causal intervention* Wang et al. [2021a], Zhang et al. [2020]. In this paper, we propose an explainable causal inference framework to adjust the confounding variables in the classification network by *front-door adjustment* Pearl [2009], Yang et al. [2021] to generate high-quality CAMs for any existing WSSS that requires them.

## 2 CAUSAL CLASS ACTIVATION MAPS

### 2.1 CLASS ACTIVATION MAP

To generate CAM Zhou et al. [2016], the first step is to train a multi-label classification network with global average pooling (GAP) layer followed by a FC prediction layer, and minimizing the binary-cross entropy (BCE) loss. Once the model converges, the CAM of class $z$ in an image $\mathbf{x}$ can be extracted by

$$\text{CAM}_z(\mathbf{x}) = \frac{\text{ReLU}(\mathcal{A}_z)}{\max(\text{ReLU}(\mathcal{A}_z))}, \ \mathcal{A}_z = \mathbf{w}_z^T f(\mathbf{x}) \quad (1)$$

where $\mathbf{w}_z$ denotes the FC weights corresponds to the $z$-class, and $f(\mathbf{x})$ denotes the feature map of $\mathbf{x}$ before GAP. For instance, for a ResNet50 He et al. [2015] trained on PASCAL VOC 2012 dataset for multi-label classification, $f(\mathbf{x}) \in \mathbb{R}^{2048 \times 32 \times 32}$ and $\mathbf{w} \in \mathbb{R}^{20 \times 2048}$.

### 2.2 STRUCTURAL CAUSAL MODEL

Our motivation is that if the quality of CAMs can be enhanced, other WSSS algorithms (such as IRN Ahn et al. [2019]) employ CAMs should expect a performance boost. In Section 2.2, we detail the Structural Causal Model (SCM) described in Figure 1(c). In Section 2.3, we introduce our method of applying front-door adjustment for a pre-trained classification network. In Section 2.4, we justify mathematically the improvement caused by C²AM from the optimization point of view.

To analyze the causality between image $x \in \mathbb{R}^{3 \times H \times W}$, image-level tag $z \in \mathbb{R}$, pixel-level localization $y \in \mathbb{R}^{1 \times H \times W}$, and a set of class-specific latent confounders $\{C_z\}$, we present a SCM as illustrated in Figure 1(c). Here, confounders can be any factors that trick the classifier to attend spurious localization via $P(Y|X)$, such as context Yang et al. [2021], Zhang et al. [2021, 2020], Shao et al. [2021], content and style Mitrovic et al. [2021]. Prior works Zhang et al. [2020], Mitrovic et al. [2021], Wang et al. [2021a] require identification and estimation of the confounders explicitly due to the demand from back-door adjustment. Conversely, the front-door adjustment does not necessitate the knowledge of the confounders Yang et al. [2021], Pearl [2009]. C²AM does not pinpoint the confounders directly, as C²AM utilized front-door adjustment for deconfouding.

Moreover, we argue it is necessary and essential to presume the existence of class-specific confounder $C_z$, since a different class of foreground objects has distinct properties and the confounder could create spurious correlation differently for each class. For instance, suppose the confounder is "context", the confounder for `horse` might associate `horse` with `person` as both objects co-occur frequently, while the confounder for `aeroplane` might correlate `aeroplane` with the unlabeled background object, such as `cloud`.

$\mathbf{C_z} \rightarrow \mathbf{X}$: This edge indicates the data generation process of an image $x$ by the class-specific confounder $C_z$, such as content and style Havaei et al. [2021], Mitrovic et al. [2021], Kazemi et al. [2019], Wang et al. [2021b], Gatys et al. [2015], as well as context Yang et al. [2021], Zhang et al. [2021, 2020], Shao et al. [2021]. The content could include various kinds of objects but still belong to the same semantic class, such as the dogs in a dataset might have different species, but they are considered as `dog`, and the style contains colours, lighting conditions and camera lens characteristics. To stimulate all possible combinations of the data generation factors, we propose to use causal intervention for this link, to pursue the true causality from image $x$ to localization cue $y$.

$\mathbf{C_z} \rightarrow \mathbf{Y_z}$: This link emphasizes the attentions $Y$ of a classifier are the effect of the class-specific confounder $C_z$. For a classification task, the confounder $C_z$ might help learn a better association between image $x$ and its label $z$, especially when training and test set are i.i.d. Wang and Jordan [2021], Wang et al. [2021a]. For instance, one commonly assumed confounder, context Yang et al. [2021], Zhang et al. [2021, 2020], Shao et al. [2021], introduces the non-causal features via $P(Y|X)$, e.g., `bird` co-occurs frequently with `tree`, and $P(Y|X)$ might mistakenly focus on `tree` features instead of `bird`.

$\mathbf{X} \rightarrow \mathbf{Z}$: This link indicates that the image-level tag $Z$ is an effect of image $X$. As the image-level tag $Z$ is determined and annotated by the dataset collector, and the objects in a

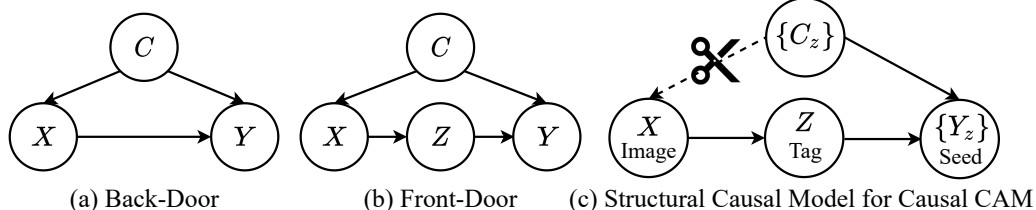



(a) Back-Door      (b) Front-Door      (c) Structural Causal Model for Causal CAM



Figure 1: Structural Causal Models (SCMs) for illustrating the fundamental (a) Back-Door model and (b) Front-Door model (c) presents our proposed SCM to analyze the generation of the localization $Y_z$, in which the image $X$, image-level tag $Z$, pixel-level localization $Y_z$, and class-specific confounder $C_z$ can be formulated in a front-door model. The "scissor" in (c) denotes causal intervention. See Section 2.2 for details.

dataset are not annotated exhaustively.

$\mathbf{Z \to Y_z}$: This edge emphasizes that a weak localization cue $Y$ is an effect of the image-level tag $z$, as the computation of CAM from the trained classifier in Eq. 1 requires an image-level label. Hence, the image-level tag $Z$ is a *mediator* Pearl [2009] that helps us estimate the causal effect of image $X$ on localization $Y_z$ for a class $z$.

## 2.3 FRONT-DOOR ADJUSTMENT FOR A CLASSIFICATION NEURAL NET

The overview of our approach is illustrated in Figure 2. For a target multi-label dataset, a standard classification neural network is first trained with BCE loss. Afterwards, a class-specific CAM of each image can be extracted by Eq. 1. The probability of an image $x \in \mathbb{R}^{3 \times H \times W}$ belonging to a class $z$ predicted by the classification network is denoted as $P(z|x)$. Further, the distribution of a CAM for an image $x \in \mathbb{R}^{3 \times H \times W}$ and a class $z$ is denoted as $P(y|x, z)$, where $y \in \mathbb{R}^{1 \times H \times W}$. In Section 2.2, we argued the assumption of the class-specific confounder $C_z$. Thus, to perform classification, we utilize the class-specific adjusted map for $z$, denoted as $P(Y_z|do(X)) = P(Z|X = x) \sum_{x_z \in X_z} P(Y|X = x_z, Z = z)P(X = x_z)$

$$P(Y|do(X)) = \sum_z P(Y_z|do(X))$$

$$= \sum_z \underbrace{\overbrace{P(z|x)}^{\text{Prob. for } z} \overbrace{\sum_{x_z \in X_z} P(Y|x_z, z)P(x_z)}^{\text{Global CAM for } z \text{ over training set}}}_{P(Y_z|do(X)): \text{ Class-specific adjusted map for } z \text{ of } x} \quad (2)$$

The derivation of Eq. 2 is shown in Appendix. To calculate $P(Y_z|do(X))$ in Eq. 2, it needs the prior knowledge of $P(X = x_z)$, which is the probability of an image $x$ belongs to class $z$ occurring. Inspired by Amrani and Bronstein [2021], we assume the training samples are equiprobable, i.e., $P(X = x_z)$ is a uniform distribution. $P(Y_z|do(X)) \in \mathbb{R}^{1 \times H \times W}$ now can be computed with the following available quantities:

- $P(Z = z|X)$: the probability of an image $x$ for class $z$ can be computed by the classifier.
- $P(X = x_z)$: assuming that each training sample is equiprobable, the probability of an image $x$ of class $z$ occurs is approximately $\frac{1}{N_z}$.
- $P(Y = y_z|X = x_z, Z = z)$: the probability distribution for the localization $y_z \in \mathbb{R}^{1 \times H \times W}$ can be computed by Eq. 1 with a trained classifier.

The semantic meaning of $\sum_{x_z \in X_z} P(Y|X = x_z, Z = z)P(X = x_z)$ in Eq. 2, is the expectation of localization $y_z$ of the entire training images for a class $z$. We term this quantity as *Global CAM*. Note, Global CAM only needs to be calculated once with the pre-trained classifier. Therefore, the computation overhead is negligible. Global CAM can be treated as a *prior* in the training set for the probability of the object for class $z$ occur in each pixel. Visualizations of Global CAMs for all classes are shown in Appendix. To train the classifier, we employ the Multiple Instance Learning technique Pinheiro and Collobert [2015], by pooling the adjusted attention map in Eq. 2 into a score $s_z \in \mathbb{R}$ for class $z$

$$s_z = \text{Pooling}(P(z|x) \sum_{x_z \in X_z} P(y_z|x_z, z)P(x_z)) \quad (3)$$

Thus, $s_z$ in Eq. 3 is treated as the prediction score to train the classification network to minimize the BCE loss. After the training is converged, the enhanced CAMs can be produced by Eq. 1 with the front-door adjusted classifier. The implementation simply requires one more line of code for a classifier training loop. See Appendix for details.

## 2.4 RELATIONS WITH OPTIMIZATION

The BCE loss for multi-label classification is defined as

$$\mathcal{L}_{bce} = -\frac{1}{Z} \sum_i \mathbf{z}_i \log \sigma(\mathbf{s}_i) + (1 - \mathbf{z}_i) \log(1 - \sigma(\mathbf{s}_i)) \quad (4)$$

where $Z$ denotes the number of classes, $\mathbf{z}$ denotes the ground-truth label, and $\mathbf{s}$ denotes the logit. The gradient

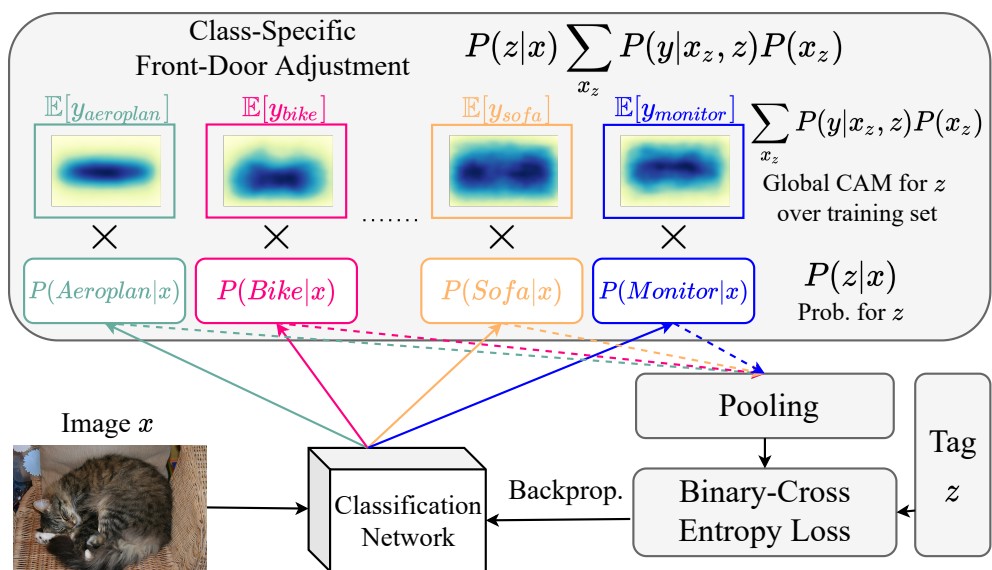

Figure 2: Overview of the proposed method of applying front-door adjustment for a classification network. See details in Section 2.3.

of $\mathcal{L}_{bce}$ w.r.t. logit $s$ can be derived as

$$\nabla_{\mathbf{s}}\mathcal{L}_{bce} = \frac{\sigma(\mathbf{s}) - \mathbf{z}}{Z} \tag{5}$$

Suppose the Pooling in Eq. 3 is a Global Average Pooling (GAP) operator. Rewrite Eq. 2 as

$$
\begin{aligned}
s_z &= \text{GAP}(P(Z=z|X=x)\textstyle\sum_{x_z \in X_z} P(Y|X=x_z, Z=z)P(X=x_z)) \\
&= P(Z=z|X=x)\text{GAP}(\textstyle\sum_{x_z \in X_z} P(Y|X=x_z, Z=z)P(X=x_z)) \\
&= P(Z=z|X=x)M_z
\end{aligned}
\tag{6}
$$

where $M_z$ is a constant computed by the GAP operator on the Global CAM for class $z$. Essentially, the logits in Eq. 4 and Eq. 5 are multiplied by a constant $M_z$.

For positive class $p$, $\mathbf{z}_p = 1$, the gradient of $\mathcal{L}_{bce}$ w.r.t. class $p$ is

$$
\begin{aligned}
\nabla_{\mathbf{s}_p}\mathcal{L}_{bce} &= -\frac{1}{Z}\nabla_{\mathbf{s}_p}(\mathbf{z}_p \log \sigma(M_p\mathbf{s}_p)) \\
&= -\frac{1}{Z}\frac{M_p}{e^{M_p\mathbf{s}_p}+1} = -M_p e^{-M_p\mathbf{s}_p}\frac{\sigma(M_p\mathbf{s}_p)}{Z}
\end{aligned}
\tag{7}
$$

For negative class $q$, $\mathbf{z}_q = 0$, the gradient of $\mathcal{L}_{bce}$ w.r.t. class $q$ is

$$
\begin{aligned}
\nabla_{\mathbf{s}_q}\mathcal{L}_{bce} &= -\frac{1}{Z}\nabla_{\mathbf{s}_q}((1-\mathbf{z}_q)\log(1-\sigma(M_q\mathbf{s}_q))) \\
&= \frac{1}{Z}\frac{M_q e^{M_q\mathbf{s}_q}}{e^{M_q\mathbf{s}_q}+1} = M_q\frac{\sigma(M_q\mathbf{s}_q)}{Z}
\end{aligned}
\tag{8}
$$

The visualizations for $\nabla_{\mathbf{s}_p}\mathcal{L}_{bce}$ w.r.t. positive aeroplane label (Eq. 7) and $\nabla_{\mathbf{s}_q}\mathcal{L}_{bce}$ w.r.t. other negative labels (Eq. 8)

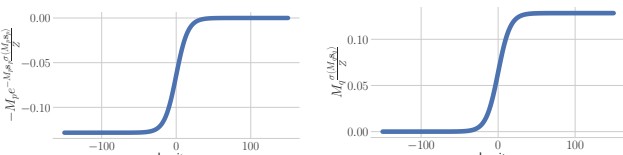

Figure 3: For illustration purpose, suppose the logit $s$ by Eq. 6 ranges from $-150$ to $150$, and the visualizations of (a) the $\nabla_s\mathcal{L}_{bce}$ w.r.t. positive aeroplane label (Eq. 7), and (b) the $\nabla_s\mathcal{L}_{bce}$ w.r.t. to other negative labels (Eq. 8). The plot contains only $M_z$ for class aeroplan in the PASCAL VOC 2012 dataset.

for the logit $s$ from $-150$ to $150$ are presented in Figure 3. From Figure 3(a), it can be deduced that for positive labels and positive logits, the gradient is nearly 0, which indicates that for correct predictions, the performance of the classifier is not impacted. However, for positive labels and negative logits, the gradient is non-zero and negative, which means that the gradient continues to push the weights of the classifier to minimize the BCE loss. Similar analysis can also be applied to Figure 3(b).

## 3 EXPERIMENTAL RESULTS

In Section 3.1, we introduce the dataset, evaluation metric, and state-of-the-art algorithms. In Section 3.2, the effectiveness of C$^2$AM is demonstrated both quantitatively and qualitatively, and the comparisons with the state-of-the-arts are reported. For reproducibility, the random seed is fixed as 0 for all experiments.

| Method | Type | Backbone | Seed | Pseudo-Mask | val | test |
|---|---|---|---|---|---|---|
| CAM Zhou et al. [2016]$_{CVPR'16}$ | / | ResNet50 | 48.3 | 65.9 | 63.5 | 64.8 |
| CONTA Zhang et al. [2020]$_{NeurIPS'20}$ | $\mathcal{A}, \mathcal{C}$ | ResNet50 | 48.8 | 67.9 | 65.3 | 66.1 |
| CONTA+SEAM Wang et al. [2020] | $\mathcal{A}, \mathcal{C}$ | ResNet38 | 56.2 | 65.4 | 66.1 | 66.7 |
| C$^2$AM (Ours) | $\mathcal{C}$ | ResNet50 | 52.1 | 69.6 | 67.5 | 67.7 |
| AdvCAM Lee et al. [2021]$_{CVPR'21}$ | $\mathcal{I}$ | ResNet50 | 55.6 | 69.9 | 68.1 | 68.0 |
| ReCAM Chen et al. [2022]$_{CVPR'22}$ | $\mathcal{A}$ | ResNet50 | 54.8 | 70.8 | 68.7 | 68.5 |
| RCA Zhou et al. [2022]$_{CVPR'22}$ | $\mathcal{M}$ | ResNet38 | / | 74.1 | 72.2 | 72.8 |

Table 1: Quantitative comparison with state-of-the-arts in mIoU (%) on the PASCAL VOC 2012 dataset. IRN Ahn et al. [2019] is the default algorithm to produce the Pseudo-Masks on the CAM seeds generated by various algorithms. The results of the prior causal framework for WSSS, CONTA, include both IRN and SEAM Wang et al. [2020]. $\mathcal{A}$ denotes using additional parameters. $\mathcal{C}$ denotes employing causal inference. $\mathcal{I}$ denotes manipulating of images. $\mathcal{M}$ denotes the utilization of a memory bank. DeepLabV2 Chen et al. [2018] is trained on the pseudo-masks, and the mIoUs of its segmentation prediction on the validation and test sets are reported.

## 3.1 SETTINGS

PASCAL VOC 2012 Everingham et al. [2010] is a commonly used dataset for evaluating semantic segmentation algorithms, it contains 20 foreground object categories and 1 background class. Following the conventional practice in related works Ahn et al. [2019], Chen et al. [2022], Zhang et al. [2020], the training set is augmented with additional data proposed by Hariharan et al. [2011]. In total, there are 10,582 images in the training set, 1,499 images in the validation set, and 1456 images in the test set.

For training our method, a ResNet50 He et al. [2015] is pre-trained on the PASCAL VOC 2012 for the multi-label classification task. Afterwards, we generate the Global CAM for each class, and the classifier is then trained by the front-door adjustment as shown in Figure 2.

## 3.2 RESULTS

### 3.2.1 Quantitative evaluation

Quantitative evaluations are shown in Table 1. Three types of masks are evaluated. First, seed area masks are produced by CAMs. Second, pseudo-masks constructed by IRN Ahn et al. [2019] based on CAM seed area. Third, segmentation masks are predicted by DeepLabV2 trained on the pseudo-masks. The standard quantitative evaluation metric, mean Intersection over Union (mIoU), was computed against the ground-truth pixel-level masks. Moreover, we compared C$^2$AM with three CAM generation algorithms, vanilla CAM Zhou et al. [2016], AdvCAM Lee et al. [2021], ReCAM Chen et al. [2022], and one causal inference algorithm, CONTA Zhang et al. [2020]. Specifically, Adv-CAM Lee et al. [2021] requires successive manipulation of images in every iteration, ReCAM Chen et al. [2022] and CONTA Zhang et al. [2020] require additional network parameters. C$^2$AM does not require any additional parameters,

network architecture changes, or manipulation of images.

As shown in Table 1, C$^2$AM outperforms the vanilla CAM (+3.7%) and the prior causal framework CONTA (+1.7%) by a large margin in the Pseudo-Mask generation section, which also improved the mIoU on validation and test set of the PASCAL VOC 2012 dataset predicted by a standard DeepLabV2 trained on pseudo-masks. Nevertheless, C$^2$AM didn't outperform the state-of-the-art, such as Advcam Lee et al. [2021], ReCAM Chen et al. [2022] and RCA Zhou et al. [2022]. We analyze the reasons in the qualitative evaluation in Section 3.2.2. Interestingly, we test the idea of multiplying a constant, such as 0.2, to the classification logits, during training the classification network, and it also outperforms the vanilla CAM for all three types of masks.

### 3.2.2 Qualitative evaluation

Qualitative evaluations are shown in Figure 4. In most cases, C$^2$AM does enhance the quality of the CAM by covering more parts of the objects, which causes the improvement of pseudo-masks. However, as shown in the last row in Figure 4, it is noticeable that C$^2$AM tends to over-localize the object, which causes the pseudo-mask contains over-segmentation areas. This is the main reason that we didn't outperform the state-of-the-arts.

## 4 CONCLUSIONS

In this work, we aim to ameliorate the quality of CAMs, which intuitively should conduce to the performance of any existing WSSS algorithms that utilize CAMs. We formulate the generation of CAM in a front-door model from causality, and we quantitatively and qualitatively demonstrated the simplicity and effectiveness of this method. C$^2$AM outperforms the vanilla CAM and prior causal framework CONTA,

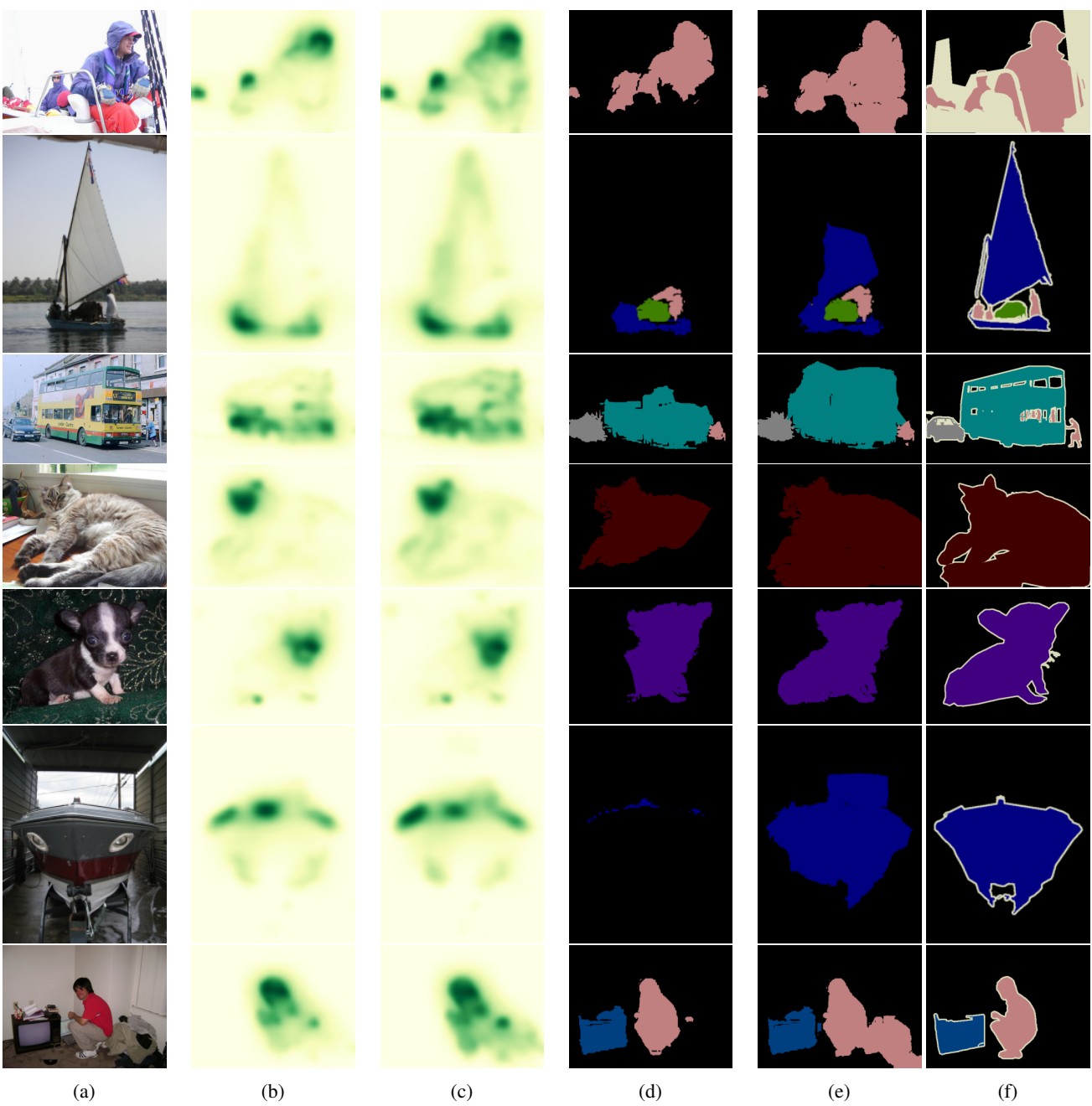

| (a) | (b) | (c) | (d) | (e) | (f) |

Figure 4: Qualitative evaluation of C²AM and its pseudo-mask generated by IRN Ahn et al. [2019] on the PASCAL VOC 2012 training set. (a) original images. (b) CAM Lin et al. [2016]. (c) C²AM. (d) pseudo-masks generated by IRN with CAMs. (e) pseudo-masks generated by IRN with C²AM. (f) pixel-level ground truth. C²AM highlights more regions of objects than CAM, which conduces to a better quality of pseudo-masks. However, C²AM tends to produce over-localized objects, which causes the IRN to over-segment the objects. The last row shows a failure case due to over-activation.

while didn't reach the state-of-the-arts. Further research is required to alleviate the over-localization issue, such as applying regularization.

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
