# OpenReview forum: "Causal Class Activation Maps for Weakly-Supervised Semantic Segmentation"
_auai.org/UAI/2022/Workshop/CRL — CRL@UAI 2022 Oral_

### Official Review · Reviewer_XqLn · 2022-06-29
**Interesting paper on CAMs inspired by causality for WSSS, could benefit from more aligned and rigorous evaluation**

**Rating:** 7
**Confidence:** 4

**Review:**

_**Contributions**_

The authors propose a new Causal Inference framework for producing Class Activation Maps (CAMs) in Weakly Supervised Semantic Segmentation (WSSS). The proposed approach aims to de-confound the classification network which produces CAMs (Class Activation Maps) via front door adjustment, hoping to produce a more robust seed for downstream pseudo ground truth pixel generation. The authors also provide an explanation from the lens of optimization for supporting their claims.

The method essentially involves multiplying the pre-trained classifier logits with a pooled score from the Global CAM for a given class and retraining the network. Assuming the presence of class specific confounders,  this global CAM score is derived to be the “class adjusted map" which accounts for the intervention of the link between the input $X$ and the attention map $Y$ i.e. $P(Y|do(X))$ for a given class.

_**Strengths**_

Overall, the paper is well organized and well presented. The proposed causal model seems to be reasonable and is novel as compared to the previous causal framework which utilises back door adjustment instead. It is nice to see an explanation of every link in the model. While evaluation, the authors also consider very recent works from venues like CVPR '22 which is appreciated. The method achieves reasonable performance given minimal overhead of parameters or implementation.

_**Main Concerns**_

1. The main concern here is whether the efficacy i.e. improvement in mIoU is actually due to enhanced causal attribution in the CAMs, or simply an artefact of multiplying the classifier logits with a numerical value?
2. There needs to be some evidence on whether the proposed method leads to CAMs which focus on the core feature rather than the spurious feature (like background - the overlocalization effect infact does incorporate extra background). In this regard, the paper needs more rigorous evaluation  - it would be nice to see some evidence (atleast qualitative) to ensure that the increase in IOU is indeed due to accurate attribution to the object in concern.
3. The authors make a vague statement on "Interestingly, we test the idea of multiplying a constant, such as 0.2, to the classification logits, during training the classification network, and it also outperforms the vanilla CAM for all three types of masks." - doesn't this further support point 1 and weaken the alignment of the results w.r.t causality?
4. How realistic is the assumption on all training samples being equiprobable? This should be addressed/motivated during the derivation.

_**Other questions and suggestions**_

It would be nice if the authors can incorporate these questions into the next iteration of the paper:

1. Doesn't figure 3 hold even if we dont multiply $M$ to the logits? How does figure 3 look like for other methods used for comparison?
2. The intervention of this particular link can be motivated more precisely. The following line is not very clear: “To stimulate all possible combinations of the data generation factors, we propose to use causal intervention for this link, to pursue the true causality from image x to localization cue y.”
3. It would be nice to have a brief description of the difference between front door and back door adjustment (in a few sentences).
4. Before explaining each link in the SCM, the SCM variables should be summarised in a single place as the figure appears later.
5. Equation 3 can be connected with equation 2 as $s = Pooling(P(Y_z|do(X)) $ to make the connection more coherent to the reader.


Note: The link to the code and the appendix could not be accessed and hasn't been considered in the above evaluation.

---

### Official Review · Reviewer_65Xo · 2022-07-05
**Nice use of causality in computer vision**

**Rating:** 7
**Confidence:** 4

**Review:**

This paper proposes to use causal tools (in particular the front-door adjustment) to improve the quality of class activation maps which are commonly used as part of many modern segmentation pipelines in vision. The proposed model is simple and the initial experimental results underscore its effectiveness. It would be nice to see more experimental evaluation on datasets which are more complex than Pascal voc2012; for example the COCO-stuff dataset (https://github.com/nightrome/cocostuff) is a good candidate as it has more complex visual scenes with multiple objects to segment. The paper is written reasonably well and it was easy to follow and understand the method, although at some points it would have been nice to have more background and/or explanations, e.g. the definition of class activation maps which is central to this paper. In summary, this is an interesting paper that should be accepted.

---

### Meta-Review · Program_Chairs · 2022-07-06

**Recommendation:** Accept (Oral)
**Confidence:** 4

**Metareview:**

Interesting application of causality in computer vision, should be discussed at the workshop.

---

### Decision · Program_Chairs · 2022-07-06

Accept (Oral)